# Local Consolidative Therapy May Have Prominent Clinical Efficacy in Patients with *EGFR*-Mutant Advanced Lung Adenocarcinoma Treated with First-Line Afatinib

**DOI:** 10.3390/cancers15072019

**Published:** 2023-03-28

**Authors:** Ming-Ju Tsai, Jen-Yu Hung, Juei-Yang Ma, Yu-Chen Tsai, Kuan-Li Wu, Mei-Hsuan Lee, Chia-Yu Kuo, Cheng-Hao Chuang, Tai-Huang Lee, Yen-Lung Lee, Chun-Ming Huang, Mei-Chiou Shen, Chih-Jen Yang, Inn-Wen Chong

**Affiliations:** 1Department of Internal Medicine, Division of Pulmonary and Critical Care Medicine, Kaohsiung Medical University Hospital, Kaohsiung Medical University, Kaohsiung 80756, Taiwan; 2School of Medicine, College of Medicine, Kaohsiung Medical University, Kaohsiung 80708, Taiwan; 3Department of Internal Medicine, Kaohsiung Municipal Ta-Tung Hospital, Kaohsiung Medical University, Kaohsiung 80145, Taiwan; 4Department of Internal Medicine, Kaohsiung Municipal Siaogang Hospital, Kaohsiung Medical University, Kaohsiung 81267, Taiwan; 5Department of Surgery, Kaohsiung Municipal Ta-Tung Hospital, Kaohsiung Medical University, Kaohsiung 80145, Taiwan; 6Department of Radiation Oncology, Kaohsiung Municipal Ta-Tung Hospital, Kaohsiung Medical University, Kaohsiung 80145, Taiwan; 7Department of Pharmacy, Kaohsiung Medical University Hospital, Kaohsiung Medical University, Kaohsiung 80756, Taiwan; 8School of Post-Baccalaureate Medicine, College of Medicine, Kaohsiung Medical University, Kaohsiung 80708, Taiwan; 9Department of Respiratory Therapy, College of Medicine, Kaohsiung Medical University, Kaohsiung 80708, Taiwan

**Keywords:** epidermal growth factor receptor, tyrosine kinase inhibitor, afatinib, local consolidative therapy

## Abstract

**Simple Summary:**

Afatinib is an irreversible epidermal growth factor receptor (EGFR) tyrosine kinase inhibitor (TKI) used to treat patients with advanced *EGFR*-mutant lung cancer. In this study, we retrospectively enrolled patients with lung adenocarcinomas harboring susceptible *EGFR* mutations, who were diagnosed and treated with first-line afatinib in three Kaohsiung Medical University-affiliated hospitals. Patients who received local consolidative therapy (LCT) had a significantly longer PFS than those who did not. Patients who received LCT also had significantly longer OS. Multivariable analysis showed LCT was an independent prognostic factor for improved PFS and OS. The analyses using propensity score-weighting showed consistent results. We conclude that LCT may improve clinical outcomes, in terms of PFS and OS, in patients with advanced *EGFR*-mutant lung adenocarcinomas who are treated with first-line afatinib. Further large-scale prospective trials are warranted.

**Abstract:**

Afatinib is an irreversible tyrosine kinase inhibitor (TKI) targeting the epidermal growth factor receptor (EGFR), which is utilized for the treatment of patients with advanced lung cancer that harbors *EGFR* mutations. No studies have evaluated the clinical efficacy of LCT in patients treated with first-line afatinib. In this study, we retrospectively enrolled patients with advanced lung adenocarcinomas harboring susceptible *EGFR* mutations who were diagnosed and treated with first-line afatinib in three hospitals. A total of 254 patients were enrolled, including 30 (12%) patients who received LCT (15 patients received definitive radiotherapy for the primary lung mass and 15 patients received curative surgery). Patients who received LCT had a significantly longer PFS than those who did not (median PFS: 32.8 vs. 14.5 months, *p* = 0.0008). Patients who received LCT had significantly longer OS than those who did not (median OS: 67.1 vs. 34.5 months, *p* = 0.0011). Multivariable analysis showed LCT was an independent prognostic factor for improved PFS (adjusted hazard ratio [aHR] [95% confidence interval (CI)]: 0.44 [0.26–0.73], *p* = 0.0016) and OS (aHR [95% CI]: 0.26 [0.12–0.54], *p* = 0.0004). The analyses using propensity score-weighting showed consistent results. We conclude that LCT may improve clinical outcomes, in terms of PFS and OS, in patients with advanced *EGFR*-mutant lung adenocarcinomas who are treated with first-line afatinib.

## 1. Introduction

Lung cancer is the leading cause of cancer-related mortality worldwide, including in Taiwan. Because most lung cancer patients are diagnosed at an advanced stage, salvage therapy is typically recommended. Platinum-based chemotherapy is the standard therapy for advanced-stage lung cancer but has modest clinical efficacy and may cause various adverse drug reactions (ADRs), such as nausea, vomiting, hematological toxicity, and unexpected, life-threatening complications [1].

Many new targeted therapies have been developed, showing better clinical efficacy when compared with standard platinum-based chemotherapy. Several large-scale, Phase III clinical trials have demonstrated better clinical outcomes in patients with lung cancer harboring susceptible epidermal growth factor receptor (*EGFR*) mutations who were treated with tyrosine kinase inhibitors (TKIs), including improvements in overall response rate (ORR), progression free survival (PFS), and quality of life, compared with those receiving platinum-based chemotherapy [2,3,4,5].

Afatinib is an irreversible, second-generation EGFR-TKI. In a head-to-head phase III randomized controlled trial, patients receiving afatinib as first-line therapy had significantly longer PFS and time to treatment failure when compared with patients receiving the first-generation EGFR-TKI, gefitinib [6]. In patients with *EGFR* exon 19 deletion, afatinib users had a significantly longer OS than chemotherapy [7]. In several retrospective, real-world studies, afatinib showed better clinical effectiveness [8,9,10,11,12,13]. In addition, a lower starting dose of afatinib demonstrated less severe ADRs than a higher dose, while achieving similar outcomes [14,15].

Therefore, afatinib is considered a promising EGFR-TKI for treating patients with *EGFR*-mutant lung cancer. Although EGFR-TKIs have been demonstrated to be associated with improved ORR and PFS in advanced *EGFR*-mutant lung adenocarcinoma more than traditional chemotherapy regimens, the development of resistance to EGFR-TKIs typically occurs within 10–14 months after treatment initiation [2,14,15,16,17,18]. The importance of developing strategies to overcome target therapy resistance cannot be overemphasized, but there remains an unmet need for non–small cell lung cancer (NSCLC) treatment.

Local consolidative therapy (LCT) for lung cancer includes curative surgical interventions and definitive radiotherapy. Many small-scale studies have investigated whether LCT can improve the outcomes for selected patients with advanced lung cancer [19,20,21,22,23]. An analysis of the Surveillance, Epidemiology, and End Results (SEER) database in 2017 suggested that multimodal therapies, particularly the addition of thoracic surgery to chemotherapy, are associated with a significantly improved prognosis [21]. The first multi-institutional randomized trial, led by the MD Anderson Cancer Center, demonstrated improved PFS and OS associated with LCT, when compared with standard maintenance therapy [24]. Iyengar et al. also demonstrated that noninvasive stereotactic ablative radiotherapy (SABR) prolonged PFS from 3.5 to 9.7 months [25]. Palma et al. reported a phase II trial (SABR-COMET trial), showing that SABR-based LCT resulted in prolonged PFS and OS in several solid tumors, including lung cancer, compared with observation only [26]. Surgical interventions also may provide extra benefits when combined with standard systemic chemotherapy in patients with advanced NSCLC [22,27,28].

Recently, several retrospective studies have indicated that LCT might provide additional clinical benefits in patients with advanced *EGFR*-mutant NSCLC who are treated with first-line EGFR-TKIs. In a retrospective study of 42 patients receiving surgery after a good response to TKI or immunotherapy, the five-year disease-free survival and overall survival rates were 76.5% and 66.0%, respectively; although this study did not enroll a control group, the encouraging outcomes still demonstrated the feasibility of surgical resection for residual tumor after TKI or immunotherapy [29]. In a study by Elamin et al., patients with *EGFR*-mutant NSCLC treated with a first-line EGFR-TKI followed by LCT had significantly longer PFS when compared with those treated with treatment with a first-line EGFR-TKI alone [19]. Several smaller-scale trials in Taiwan also demonstrated that patients receiving an EGFR-TKI together with local surgical resection to treat primary lung tumors also experienced significantly longer PFS than those receiving an EGFR-TKI alone [22,27,28]. A meta-analysis of these available data showed that LCT might improve PFS and OS in metastatic NSCLC patients, without increasing the risks of high-grade adverse effects [23].

These studies indicate that the combination of EGFR-TKIs with LCT may represent a promising strategy for advanced *EGFR*-mutant lung cancer. However, it has not been investigated as to whether the irreversible EGFR-TKI afatinib plus LCT is associated with improved outcomes compared with afatinib alone.

We conducted a multicenter, retrospective cohort study to investigate the impacts of adding LCT on targeting the primary tumor mass in patients with susceptible *EGFR*-mutant lung adenocarcinoma treated with first-line afatinib.

## 2. Materials and Methods

### 2.1. Patient Identification

We enrolled patients who were diagnosed with lung adenocarcinoma and treated with afatinib in three Kaohsiung Medical University-affiliated hospitals, including Kaohsiung Medical University Hospital, Kaohsiung Municipal Ta-Tung Hospital, and Kaohsiung Municipal Siaogang Hospital. Lung adenocarcinoma was pathologically confirmed according to the World Health Organization pathological classification. Tumor staging was assessed according to the 8th version of the American Joint Committee on Cancer staging system and confirmed by a lung cancer team. Genomic DNA was extracted from the tissue block, and real-time PCR (Cobas EGFR Mutation Test v2) was performed on genotype *EGFR* exons 18 to 21 using a ready-to-use kit, for the detection of 42 somatic *EGFR* mutations, using the Cobas z480 instrument with the Cobas 4800 analyzer. The examination techniques were consistent with our previous studies [14,15,30,31,32].

In the current study, we enrolled all patients with advanced (stage IIIB–IV) lung adenocarcinoma, who were naïve to systemic treatment and were treated with afatinib as their first-line systemic treatment (Figure 1). Only those with *EGFR* exon 19 deletion or exon 21 L858R point mutations were included, and those with potentially treatment-resistant mutations were excluded. Baseline clinical characteristics were determined by retrospective chart review, including age at diagnosis, sex, Eastern Cooperative Oncology Group (ECOG) performance status (PS), smoking history, family history, EGFR mutation, tumor–node–metastasis (TNM) status, and the presence of metastatic sites at the time of afatinib initiation. Patients who received LCT, including operation (curative surgical intervention, such as wedge resection, segmentectomy, lobectomy, or pneumonectomy) and local radiotherapy (definitive radiotherapy for the primary lung mass), were identified.

The initial treatment response was classified based on serial imaging studies, using the revised Response Evaluation Criteria in Solid Tumors (RECIST 1.1) criteria. PFS and OS were defined as the time from afatinib initiation to the date of disease progression, as observed on imaging examination, and to the date of death, respectively.

### 2.2. Statistical Analysis

Categorical and continuous variables were compared using the Chi-squared test and Student’s *t*-test, respectively. Survival times were estimated with the Kaplan–Meier method, and differences between groups were compared using the log-rank test. Cox regression analyses were used to identify predictive factors for PFS and OS. Both univariate and multivariable analyses were performed. Using a backward variable selection method, keeping only variables with *p*-values < 0.15, we developed reduced multivariable models to avoid over-adjustment. Hazard ratios (HRs) with 95% confidence intervals (CIs) are presented, in order to identify predictive factors for PFS or OS.

To balance confounding factors between groups, we adopted propensity score (PS) weighting methods, which enabled us to avoid discarding too many samples [13]. Three common PS weighting methods, including inverse probability weighting (IPW), the standardized mortality ratio weighting (SMRW), and matching weighting (MW), were evaluated. The targets of inference included the average treatment effect (ATE) in the whole population, average treatment effect among the treated population (ATT), and ATE in a subset of IPW, SMRW, and MW, respectively. The weights were computed from the generalized PS, using the multinomial logistic regression, using the LCT group as the outcome variable and the potential confounding variables, including age, sex, smoking history, ECOG PS, and various metastasis sites, as the covariates. The performance of different PS weighting methods in terms of balancing confounding factors was assessed by using kernel density plots, standardized mean differences (SMD), and effective sample size (ESS) [33]. We then selected the most appropriate weighting method, applying the weights in the statistical analyses.

All statistical analyses were performed with SAS software (version 9.4 for Windows, SAS Institute Inc., Cary, NC, USA). Statistical significance was set at a two-tailed *p* < 0.05.

## 3. Results

We identified 296 patients with lung adenocarcinoma who were treated with afatinib as their first-line systemic treatment (Figure 1), and 271 patients with advanced stage (stage IIIB–IV) cancer were included in this study.

After excluding those with *EGFR* mutations in exon 18 or 20 or with known resistance mutations, the remaining 254 patients with exon 19 deletion or exon 21 L858R point mutations were enrolled. In the study population (Table 1), 30 (12%) patients had received LCT, including 15 patients who received definitive radiotherapy for the primary lung mass and 15 patients who had received curative surgery (four, two, and nine patients received wedge resection, segmentectomy, and lobectomy, respectively).

The mean (±standard deviation) age was 64.3 ± 10.1 years, and 157 (62%) patients were women. No significant differences in mean age, sex distribution, smoking history, family history, or proportions of stage IV disease were observed between patients treated with and without LCT. The median interval between afatinib initiation and local radiotherapy was 10.3 months. Perioperative afatinib was initiated in seven patients, and the median interval between initiation of afatinib and operation in the remaining eight patients was 8.3 months. All patients treated with LCT had an ECOG PS value of less than one, while 33 (15%) patients without LCT had an ECOG PS value greater than two. Patients treated with LCT had significantly fewer metastatic sites, and a smaller proportion of brain metastases and pleural metastases/effusion than those treated with afatinib alone.

The initial treatment response was similar in patients with or without LCT (Table A1 in Appendix A). Patients with LCT had a significantly better PFS than those without LCT (median PFS: 32.8 vs. 14.5 months, *p* = 0.0008; Figure 2a). More than half of the patients who received operations did not present disease progression, and those treated with radiotherapy had a median PFS of 20.4 months (Figure A1a in Appendix A). Univariate Cox regression analyses revealed that LCT was associated with significantly better PFS (HR [95% CI]: 0.43 [0.26–0.72], *p* = 0.0011; Table 2). Worse ECOG PS, brain metastasis, pleural metastasis/effusion, bone metastasis, and liver metastasis were significantly associated with poorer PFS. Multivariable analysis using backward variable selection confirmed LCT as an independent prognostic factor for better PFS (HR [95% CI]: 0.45 [0.27–0.75], *p* = 0.0021 in model 1R; 0.44 [0.26–0.73], *p* = 0.0016 in model 2R). We also identified several independent prognostic factors for worse PFS, including pleural metastases/effusion and bone metastases (Table 2).

Patients with LCT had a significantly better OS than those without LCT (median OS: 67.1 vs. 34.5 months, *p* = 0.0011; Figure 2B). No patients who received operations have died to date, and patients who received radiotherapy had a median OS of 53.2 months (Figure A1b in Appendix A). Univariate Cox regression analyses revealed that LCT was associated with significantly better OS (HR [95% CI]: 0.32 [0.16–0.66], *p* = 0.0019; Table 3).

The male sex, any smoking history, worse ECOG PS, brain metastases, pleural metastases/effusion, and bone metastases were factors that were significantly associated with a poorer OS. Multivariable analysis with backward variable selection confirmed that LCT was an independent prognostic factor for better OS (HR [95% CI]: 0.35 [0.17–0.72], *p* = 0.0042 in model 3R; 0.26 [0.12–0.54], *p* = 0.0004 in model 4R). We also identified several independent prognostic factors for worse OS, including an elderly age, smoking history, worse ECOG PS, pleural metastases/effusion, and bone metastases (Table 3).

To identify an appropriate weighting for balancing groups, we used multinomial logistic regression with potential confounders, in order to compute the generalized PS. We then computed the weighted generalized PS based on IPW, SMRW, and MW. The kernel density plots of generalized PS distributions for the groups are shown in Figure A2 in Appendix A. All PS weighting methods reduced the ESS and SMD of covariates (Table A2 in Appendix A). Reaching a markedly reduced absolute SMD of covariates, MW was selected for the weights of the following analyses. The comparison of baseline characteristics after MW is shown in Table A3 in Appendix A. The Kaplan–Meier curves with MW showed that patients with LCT, compared with those without LCT, had a better PFS (median PFS: 32.8 vs. 15.6 months, *p* = 0.0277; Figure A3a in Appendix A) and OS (median OS: 67.1 vs. 44.6 months, *p* = 0.0475; Figure A3b in Appendix A). In Cox regression models with MW, LCT remained an independent, and better, prognostic factor for PFS (weighted HR [95% CI]: 0.51 [0.27–0.97], *p* = 0.0407) and OS (weighted HR [95% CI]: 0.41 [0.17–0.97], *p* = 0.0429) (Table A2 in Appendix A).

## 4. Discussion

In this multicenter, retrospective cohort study, we found that patients treated with first-line afatinib for advanced lung adenocarcinoma harboring susceptible *EGFR* mutations benefited from add-on LCT, leading to better PFS and OS. In multivariable analysis, LCT was an independent prognostic factor for improved PFS and OS. The analyses with propensity score-weighting showed consistent results.

Afatinib is a promising EGFR-TKI therapy for those with advanced lung adenocarcinomas harboring susceptible *EGFR* mutations. Several prospective and retrospective studies examining afatinib have demonstrated excellent response rates and PFS [16,18]. Afatinib demonstrated better efficacy as a first-line, targeted therapy than gefitinib in the Lux-Lung 7 study of patients with advanced *EGFR*-mutant lung adenocarcinomas [6]. Therefore, we enrolled all patients with advanced *EGFR*-mutant NSCLC treated with first-line afatinib in three hospitals in this study. LCT includes both local radiotherapy and surgical resection, and several studies have demonstrated the benefits of LCT for the treatment of *EGFR*-mutant NSCLC. The application of local radiotherapy may eradicate residual lung tumor cells, which are likely to eventually become refractory to EGFR-TKIs [20,24,26]. In addition, radiotherapy may produce an abscopal effect on metastatic tumors [34]. Local radiotherapy delivered to the primary NSCLC tumor can include traditional intensity-modulated radiation therapy (IMRT) [35] or stereotactic ablative radiotherapy (SABR) [25,26]. Using IMRT as LCT in the treatment of advanced NSCLC has been associated with a survival benefit, and Li et al. reported on IMRT use in 265 patients, with a median PFS and OS of 12.7 and 38.6 months, respectively [35]. In addition, SABR applied to the primary lung mass has shown efficacy in treating patients with oligometastatic NSCLC. Elamin et al. reported that patients treated with LCT (11 of 12 patients received SABR) plus EGFR-TKI had significantly longer PFS than patients treated with EGFR-TKI alone (median PFS: 36 vs. 14 months; HR [95% CI]: 0.29 [0.23–0.70], log-rank *p* = 0.0024). These findings indicate that local IMRT and SABR are both effective for the treatment of advanced NSCLC. In our cohort, patients receiving afatinib with local IMRT, compared with those receiving afatinib alone, appeared to have a trend toward better PFS (median PFS: 20.4 vs. 14.5 months) and OS (median OS: 53.2 vs. 34.5 months), but statistical significance was not reached due to the small sample size.

This study focuses on investigating the effectiveness of LCT on the primary tumor in patients treated with afatinib. Various local therapy strategies for the treatment of metastatic lesions are available in clinical practice. For example, we sometimes use local radiofrequency ablation or radiation therapy for liver metastases, but this is not the standard approach. Radiotherapy and operation are seldom applied for bone metastases, and it is not feasible to treat each bone metastasis site with local therapy. Local therapy is not applied to treat pleural effusion/metastases. Therefore, local therapy strategies for metastatic lesions are quite individualized, and the decision to initiate them depends on the doctors’ clinical judgment and the patients’ condition. It is quite difficult to analyze the effectiveness of various local therapy strategies for metastatic lesions.

Generally speaking, the majority of tumor recurrences in patients treated with an EGFR-TKI occur at the primary site. A larger tumor burden and higher heterogeneity of the primary lung tumor increase the likelihood of inducing acquired resistance. While EGFR-TKI can reduce tumor burden, it cannot completely eradicate the tumor. Residual tumor cells may contain cancer stem cells or cells with acquired resistant mutations, which may contribute to tumor progression and EGFR-TKI ineffectiveness over time. Considering the above reasons, LCT may help overcome these resistant mechanisms and lead to improvements in PFS and OS.

Several retrospective studies conducted in Taiwan have supported these hypotheses. Tseng et al. reported that patients with *EGFR*-mutant lung cancer treated with primary tumor resection, in addition to a first-line EGFR-TKI, experienced significantly longer PFS (25.1 months [95% CI: 16.6–33.7 months] vs. 9.4 months [95% CI: 8.4–10.4 months]; adjusted HR [95% CI]: 0.40 [0.30–0.54], *p* < 0.001) and OS (56.8 months [95% CI: 36.3–77.2 months] vs. 31.8 months [95% CI: 28.2–35.4 months]; adjusted HR [95% CI]: 0.57 [0.39–0.84], *p* = 0.004) than patients treated with an EGFR-TKI alone [22]. Kuo et al. reported that the median PFS increased to 29.6 (18.9–40.3) months in patients receiving an EGFR-TKI plus primary tumor resection, which was significantly longer than those who received EGFR-TKI alone (13.0 [11.8–14.2] months; *p* < 0.001) [28]. Chen et al. enrolled a total of 29 patients with *EGFR*-mutant advanced lung adenocarcinomas who also underwent surgical resection; the median PFS after surgery was 36.4 months, and the median OS was still not reached [27]. In Korea, Lim et al. conducted a study supported by the Korean Central Cancer Registry and the Lung Cancer Registration Committee, which showed that patients who underwent LCT had significantly better OS than those who did not (HR [95% CI]: 0.448 [0.242–0.829], *p* = 0.011) [36]. In China, Kuo et al. examined patients with NSCLC associated with driver gene mutations who were treated with targeted therapy, followed by salvage therapy, resulting in significantly longer PFS than in those treated with targeted therapy alone (median PFS: 23.4 vs. 12.9 months, *p* = 0.0004) [37]. In our cohort, more than half of the patients who received afatinib with OP did not die or show disease progression, suggesting the promising effects of add-on OP. Pooling the patients receiving afatinib with RT or OP together (LCT group), and comparing them with those receiving afatinib alone (no LCT group), we found the significant survival benefits of LCT in terms of PFS and OS. Multivariable analyses using backward variable selection confirmed that LCT was an independent prognostic factor for improved PFS and OS. In addition, we identified several independent prognostic factors for worse PFS, including pleural metastases/effusion and bone metastases, as well as several independent prognostic factors for worse OS, including elder age, smoking history, worse ECOG PS, pleural metastases/effusion, and bone metastases.

Our study is one of the largest-scale multicenter retrospective studies examining EGFR-TKI plus LCT for the treatment of advanced lung adenocarcinoma patients harboring susceptible *EGFR* mutations. Unlike previous studies, our study investigated the clinical effectiveness of LCT in an afatinib-based cohort rather than in a heterogenous group of patients treated with various EGFR-TKIs.

The present study had several limitations. Firstly, this was a retrospective cohort study. The decision to perform LCT depends on each physician’s judgment and the consensus reached by the physicians and patients, which may introduce selection bias. Further prospective studies are, therefore, urgently needed. Secondly, the sample size was limited. However, our study focused only on patients treated with afatinib, and this study was one of the largest studies examining the effectiveness of LCT in patients with advanced lung cancer receiving an EGFR-TKI. Thirdly, the surgical intervention group included those who underwent initial resection and those who had a residual mass after EGFR-TKI administration. Due to the small case numbers, we were unable to compare outcomes between these two groups. Fourthly, all of our patients received IMRT as RT. When compared with SABR, IMRT has been associated with abnormal respiratory function tests, alterations in functional and structural parameters, and a high incidence of radiation pneumonitis. Whether these side effects affect survival outcomes warrants further studies.

## 5. Conclusions

In conclusion, in this multicenter, retrospective cohort study, LCT was found to prolong PFS and OS in patients with advanced lung adenocarcinomas, who were harboring susceptible *EGFR* mutations and treated with first-line afatinib. Further large-scale prospective trials are warranted to confirm our findings.

## Figures and Tables

**Figure 1 cancers-15-02019-f001:**
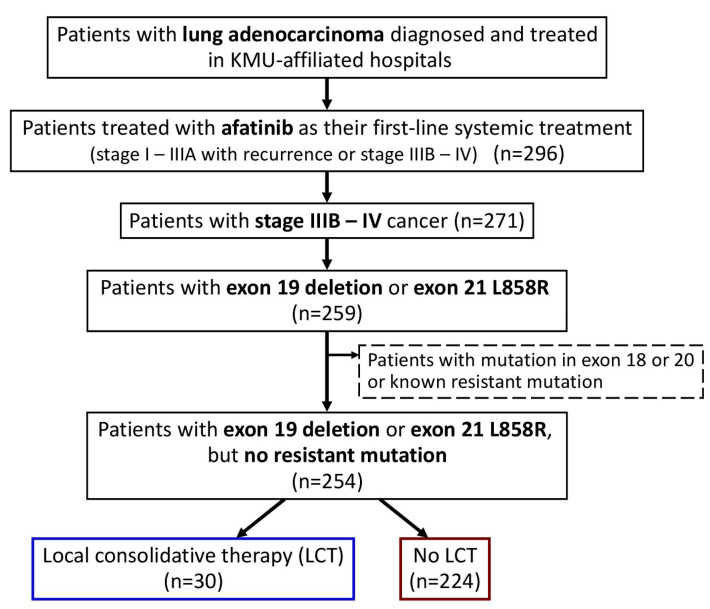
Flowchart for the identification of the study population. Abbreviation: LCT, local consolidative therapy; KMU, Kaohsiung Medical University.

**Figure 2 cancers-15-02019-f002:**
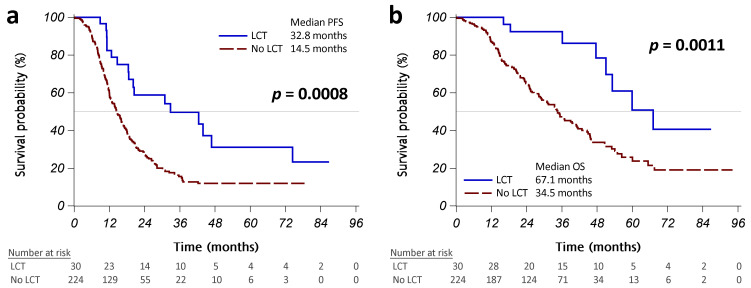
Kaplan–Meier curves, showing that progression-free survival (PFS) (**a**) and overall survival (OS) (**b**) were significantly different between the patients who received LCT (operation or radiotherapy, solid blue line) and those who did not (dashed red line). Abbreviation: LCT, local consolidative therapy.

**Table 1 cancers-15-02019-t001:** Baseline characteristics of the study cohort.

Variables	All Patients	LCT	No LCT	*p*-Value
n	254	30	224	
LCT				
No	224 (88%)	0 (0%)	224 (100%)	<0.0001
Radiotherapy	15 (6%)	15 (50%)	0 (0%)	
Operation	15 (6%)	15 (50%)	0 (0%)	
Age(year)	64.3 ± 10.1	64.7 ± 9.2	64.3 ± 10.2	0.8129
<65	128 (50%)	12 (40%)	116 (52%)	0.2253
≥65	126 (50%)	18 (60%)	108 (48%)	
Sex				
Female	157 (62%)	21 (70%)	136 (61%)	0.3256
Male	97 (38%)	9 (30%)	88 (39%)	
Smoking				
Never	202 (80%)	26 (87%)	176 (79%)	0.3021
Ever	52 (20%)	4 (13%)	48 (21%)	
Familyhistory				
No	237 (93%)	27 (90%)	210 (94%)	0.4402
Yes	17 (7%)	3 (10%)	14 (6%)	
ECOG performance status				
0–1	221 (87%)	30 (100%)	191 (85%)	0.0242
≥2	33 (13%)	0 (0%)	33 (15%)	
EGFR mutation				
Exon 19 deletion	135 (53%)	19 (63%)	116 (52%)	0.2339
Exon 21 L858R	121 (48%)	11 (37%)	110 (49%)	0.2001
Stage				
IIIB–IIIC	11 (4%)	3 (10%)	8 (4%)	0.1043
IV	243 (96%)	27 (90%)	216 (96%)	
Metastatic sites				
0–1	94 (37%)	19 (63%)	75 (33%)	0.0015
≥2	160 (63%)	11 (37%)	149 (67%)	
Metastasis				
Brain metastasis	73 (29%)	2 (7%)	71 (32%)	0.0044
Lung metastasis	133 (52%)	13 (43%)	120 (54%)	0.2917
Pleural metastasis/effusion	113 (44%)	8 (27%)	105 (47%)	0.0365
Bone metastasis	130 (51%)	14 (47%)	116 (52%)	0.5984
Liver metastasis	32 (13%)	1 (3%)	31 (14%)	0.1034
Adrenal metastasis	20 (8%)	1 (3%)	19 (8%)	0.3255
Other metastasis	18 (7%)	1 (3%)	17 (8%)	0.3936

Data are presented in n (%) or mean ± standard deviation. *p*-values were assessed with the Chi-squared test or Student’s *t*-test. Abbreviations: LCT, local consolidative therapy; EGFR, epidermal growth factor receptor; ECOG, Eastern Cooperative Oncology Group.

**Table 2 cancers-15-02019-t002:** Factors associated with progression-free survival (PFS).

Variables	Univariate	Multivariable Model 1	Multivariable Model 1R	Multivariable Model 2	Multivariable Model 2R
Local consolidative therapy (vs. no)	0.43 [0.26-0.72] **	0.46 [0.27–0.76] **	0.45 [0.27–0.75] **	0.44 [0.26–0.74] **	0.44 [0.26–0.73] **
Male (vs. female)	1.23 [0.92–1.63]	1.26 [0.90–1.77]		1.29 [0.92–1.81]	
Age (≥65 vs. <65)	0.83 [0.63–1.10]	0.85 [0.64–1.13]		0.89 [0.67–1.19]	
Smoking (ever vs. never)	1.10 [0.78–1.55]	1.01 [0.67–1.52]		0.86 [0.57–1.30]	
ECOG performance status (≥2 vs. ≤1)	1.55 [1.04–2.30] *	1.35 [0.90–2.03]	1.39 [0.94–2.07]	1.27 [0.84–1.93]	1.35 [0.90–2.01]
L858R vs. exon 19 del	1.04 [0.78–1.38]	0.97 [0.73–1.30]		1.02 [0.76–1.37]	
Stage IV (vs. stage IIIB-IIIC)	2.19 [0.81–5.90]	2.10 [0.77–5.72]			
Brain metastasis (vs. no)	1.38 [1.02–1.87] *			1.09 [0.79–1.50]	
Lung metastasis (vs. no)	0.91 [0.69–1.21]			1.00 [0.75–1.33]	
Pleural metastasis/effusion (vs. no)	1.50 [1.13–1.99] **			1.50 [1.12–2.02] **	1.49 [1.12–1.99] **
Bone metastasis (vs. no)	1.92 [1.45–2.56] ***			2.00 [1.46–2.74] ***	2.03 [1.50–2.73] ***
Liver metastasis (vs. no)	1.74 [1.18–2.56] **			1.43 [0.94–2.16]	1.34 [0.90–2.01]
Adrenal metastasis (vs. no)	1.31 [0.80–2.16]			0.96 [0.57–1.64]	
Other metastasis (vs. no)	0.87 [0.49–1.53]			0.81 [0.45–1.47]	

Data are presented as hazard ratios [95% confidence interval]. * *p* < 0.05; ** *p* < 0.01; *** *p* < 0.0001. After building the maximal models of the multivariable Cox regression (Models 1 and 2), the corresponding reduced models (Models 1R and 2R, respectively) were built using the backward variable selection method, keeping only the variables with *p*-values less than 0.15. Abbreviations: ECOG, Eastern Cooperative Oncology Group.

**Table 3 cancers-15-02019-t003:** Factors associated with overall survival (OS).

Variables	Univariate	Multivariable Model 3	Multivariable Model 3R	Multivariable Model 4	Multivariable Model 4R
Local consolidative therapy (vs. no)	0.32 [0.16–0.66] **	0.33 [0.16–0.70] **	0.35 [0.17–0.72] **	0.26 [0.12–0.56] **	0.26 [0.12–0.54] **
Male (vs. female)	1.67 [1.19–2.35] **	1.32 [0.88–2.00]		1.38 [0.92–2.06]	1.36 [0.92–2.03]
Age (≥65 vs. <65)	1.35 [0.97–1.89]	1.39 [0.99–1.95]	1.40 [1.00–1.96]	1.49 [1.06–2.11] *	1.45 [1.04–2.04] *
Smoking (ever vs. never)	1.72 [1.17–2.52] **	1.76 [1.10–2.83] *	2.03 [1.37–3.00] **	1.51 [0.96–2.37]	1.51 [0.96–2.37]
ECOG performance status (≥2 vs. ≤1)	2.69 [1.75–4.12] ***	2.53 [1.61–3.97] ***	2.70 [1.74–4.21] ***	2.49 [1.56–3.99] ***	2.62 [1.67–4.12] ***
L858R vs. exon 19 del	1.16 [0.83–1.62]	0.99 [0.70–1.40]		0.93 [0.65–1.33]	
Stage IV (vs. stage IIIB-IIIC)	1.31 [0.49–3.55]	1.55 [0.56–4.30]			
Brain metastasis (vs. no)	1.63 [1.14–2.33] **			1.21 [0.82–1.77]	
Lung metastasis (vs. no)	0.79 [0.56–1.10]			0.91 [0.64–1.29]	
Pleural metastasis/effusion (vs. no)	1.43 [1.02–2.00] *			1.42 [1.00–2.01] *	1.41 [1.00–1.98] *
Bone metastasis (vs. no)	2.21 [1.56–3.13] ***			2.62 [1.78–3.84] ***	2.74 [1.89–3.95] ***
Liver metastasis (vs. no)	1.43 [0.88–2.33]			1.23 [0.73–2.06]	
Adrenal metastasis (vs. no)	0.87 [0.46–1.66]			0.52 [0.26–1.03]	0.54 [0.27–1.05]
Other metastasis (vs. no)	0.98 [0.48–2.02]			1.16 [0.55–2.46]	

Data are presented as hazard ratios [95% confidence interval]. * *p* < 0.05; ** *p* < 0.01; *** *p* < 0.0001. After building the maximal models of the multivariable Cox regression (Models 3 and 4), the corresponding reduced models (Models 3R and 4R, respectively) were built using the backward variable selection method, keeping only the variables with *p*-values less than 0.15. Abbreviations: ECOG, Eastern Cooperative Oncology Group.

## Data Availability

The data presented in this study are available in this article.

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
