# Peer review of "Local Consolidative Therapy May Have Prominent Clinical Efficacy in Patients with EGFR-Mutant Advanced Lung Adenocarcinoma Treated with First-Line Afatinib"

_cancers, 2023, doi:10.3390/cancers15072019_

Round 1
Reviewer 1 Report
Dear Authors,
The presentation of the work is well and all studies sounds good. However, I did not found your study very novel to be published in this journal. I will suggest you publish this study in a more relevant journal for example, any clinical study or pharmacy practice journals.
Author Response
Thank you for your kind comments. Previous studies on local consolidation therapy (LCT) always included all EGFR TKIs together. However, different EGFR TKIs may vary in efficacy. This study is the first to enroll only patients receiving a single EGFR TKI, afatinib, to elucidate the effect of LCT. We believe that this study's focus on a single EGFR TKI may provide more robust clinical evidence for the combination treatment of LCT and EGFR TKI in treating lung adenocarcinoma with susceptible EGFR mutation.
Reviewer 2 Report
This research study is novel in that it looked specifically at the clinical efficacy of LCT using only one type of EGFR TKI, Afatinib. The study population was well-defined, and the inclusion and exclusion criteria were clearly described. The statistical analysis used were adequate and clearly described. The paper is clearly-written and unbiased.
The following are minor revisions:
The authors gave ‘very brief’ potential mechanisms that may explain the benefit from LCT in terms of PFS and OS. It would be nice to add more discussion on this section.
The sentence used in Line 28-29 is the same one in Line 39-40. It would be nice to paraphrase.
All the figures/graphs are quite distorted. Please check the formatting.
Author Response
Authors’ reply:
The point raised by the reviewer is important. Generally speaking, the majority of tumor recurrences in patients treated with an EGFR TKI occur at the primary site. A larger tumor burden and higher heterogeneity of the primary lung tumor increase the likelihood of inducing acquired resistance. While EGFR TKI can reduce tumor burden, it cannot completely eradicate the tumor. Residual tumor cells may contain cancer stem cells or cells with acquired resistant mutation, which may contribute to tumor progression and EGFR TKI ineffectiveness over time. Considering the above reasons, LCT may help overcome these resistant mechanisms and lead to improvements in progression-free survival (PFS) and overall survival (OS). We have included the information in the discussion of our revised manuscript.
The sentence used in Line 28-29 is the same one in Line 39-40. It would be nice to paraphrase.
Authors’ reply:
Thanks for the suggestion. We have rephrased the sentence in line 39-40 to “Afatinib is an irreversible tyrosine kinase inhibitor (TKI) targeting epidermal growth factor receptor (EGFR), which is utilized for the treatment of patients with advanced lung cancer that harbors EGFR mutations.” in our revised manuscript.
All the figures/graphs are quite distorted. Please check the formatting.
Authors’ reply:
Thanks for the careful review. We have reformatted the figures in our revised manuscript.
Reviewer 3 Report
The authors conducted a retrospective study to assess the benefits of Local consolidative therapy (LCT) in EGFR-mutant advanced lung adenocarcinoma patients treated with first-line afatinib. The analysis suggest LCT combined afatinib treatment could bring significant survival benefits, result in both longer PFS and longer OS than afatinib treatment only.
The study conception is straightforward as several similar studies have been done with other TKIs. The sample size is small in either radiotherapy LCT group or operation LCT group. I would suggest to add "Number at risk" in KM curve plots. In Fig S1A and 1B, it is not clear which groups the p value is calculated against, since there are three groups in the plot.
Author Response
Authors’ reply:
We appreciate the comments from the reviewer. As suggested, the “number at risk” are added below the KM plots in our revised manuscript. In Figure S1 (A1 in our revised manuscript), the p value <0.05 described the presence of significant differences between three groups. Because the sample sizes are quite small, we believe it is not quite suitable to emphasize the differences between any two groups in this figure. In our cohort, more than half of patients receiving afatinib with OP did not have disease progression or die yet; patients receiving afatinib with RT, compared with those receiving afatinib alone, appeared to have a trend toward better PFS (median PFS: 20.4 vs. 14.5 months) and OS (median OS: 53.2 vs. 34.5 months), but statistical significance was not reached due to small sample size. We have discussed this issue in the Discussion of the revised manuscript.